# Identifying and Understanding the Non-Clinical Impacts of Delayed or Cancelled Surgery in Order to Inform Prioritisation Processes: A Scoping Review

**DOI:** 10.3390/ijerph19095542

**Published:** 2022-05-03

**Authors:** Kathryn Jack, Catrin Evans, Louise Bramley, Joanne Cooper, Tracy Keane, Marie Cope, Elizabeth Hendron

**Affiliations:** 1Surgical Division, Nottingham University Hospitals NHS Trust, Nottinghamshire NG5 1PB, UK; louise.bramley@nuh.nhs.uk (L.B.); joanne.cooper3@nuh.nhs.uk (J.C.); tracy.keane@nuh.nhs.uk (T.K.); marie.cope@nuh.nhs.uk (M.C.); elizabeth.hendron@nuh.nhs.uk (E.H.); 2School of Health Sciences, University of Nottingham, Nottinghamshire NG7 2RD, UK; catrin.evans@nottingham.ac.uk

**Keywords:** delayed surgery, COVID-19, waiting for surgery, elective surgery, surgical cancellation

## Abstract

The COVID-19 pandemic has resulted in significant delays to non-urgent elective surgery. Decision making regarding prioritisation for surgery is currently informed primarily by clinical urgency. The ways in which decision making should also consider potential social and economic harm arising from surgical delay are currently unclear. This scoping review aimed to identify evidence related to (i) the nature and prevalence of social and economic harm experienced by patients associated with delayed surgery, and (ii) any patient assessment tools that could measure the extent of, or predict, such social and economic harm. A rapid scoping review was undertaken following JBI methodological guidance. The following databases were searched in October 2020: AMED; BNI; CINAHL; EMBASE; EMCARE; HMIC; Medline; PsychINFO, Cochrane, and the JBI. A total of 21 publications were included. The findings were categorised into five themes: (i) employment, (ii) social function and leisure, (iii) finances, (iv) patients’ experiences of waiting, and (v) assessment tools that could inform decision making. The findings suggest that, for some patients, waiting for surgery can include significant social, economic, and emotional hardship. Few validated assessment tools exist. There is an urgent need for more research on patients’ experiences of surgical delay in order to inform a more holistic process of prioritising people on surgical waiting lists in the COVID-19 pandemic recovery stages.

## 1. Introduction

This review was prompted by conditions related to the emergence of the SARS-CoV-2 (COVID-19) pandemic in the UK during January 2020 which rapidly resulted in a severe reduction in the capacity of National Health Service (NHS) Trusts to provide a timely elective surgical service. In March 2020, NHS England instructed that all elective surgery should cease by 15 April 2020 for a minimum of 3 months [1]. The NHS constitution standard decrees that 92% of people should not have to wait for non-emergency treatment for more than 18 weeks [2]. As a consequence of halting elective surgery, by the end of October 2021, 65.6% of people were still waiting at 18 weeks, a sharp deviation from the 8% deemed acceptable [2]. The number of people in England waiting at the end of October 2021 was 6 million, of whom 312,665 had been waiting in excess of 52 weeks [2]. In 2013/14, NHS England introduced a zero-tolerance policy of waiting lists being of this duration [2].

People waiting for operations are stratified according to clinical need (e.g., disease severity and/or pain). The Royal College of Surgeons (RCS) has developed guidance to determine surgical priority (with the exception of obstetrics, gynaecology and ophthalmology) during the COVID-19 pandemic [3]. It is important to note there is a distinction between “urgency” (a characteristic assigned to a patient about the speed required in order to obtain or maximise the desired outcome) and “priority” (the patients position relative to others on the waiting list [4]. Table 1 summarises the definitions of the RCS priority level classifications.

Prior to and during the pandemic, the impact of delayed surgical procedures on individual non-clinical, or non-physical harms has been an area of significant concern. There are reports of profound social harms, such as loss of earnings due to being unable to work, relationship breakdown, and difficulties in obtaining assistance with activities of daily living [5]. Delays experienced by patients can also impact nursing care provision. If people are more clinically unwell or have experienced some loss in their ability to self-care, this may change their in-patient nursing needs and require additional staffing resource and changes in skill-mix. Thus, there is an emerging need to consider stratifying peoples’ waiting list position within the RCS surgical priority category to which they have been assigned, based not just on potential physical harms resulting from an extended delay in resolving their clinical condition, but also risk of non-clinical harms.

Against this background, a rapid scoping literature review was undertaken to understand, through a more holistic lens, the social difficulties that may arise from an extended wait for surgery. Anecdotal local evidence indicates increasing social effects of extensive waiting times for surgery, including significant distress to patients and their families. This, in turn, may impact on moral injury and subsequent psychological ill-health on nursing, care staff and administration and clerical teams. The aim of this rapid scoping review is to identify material to inform the construction of a contemporary assessment tool to aid surgical priority decision making in the aftermath of the COVID-19 pandemic. A pragmatic, problem-solving approach was considered necessary by the research team to gather information to use to improve patient care. The literature review addresses the following two questions:What are the non-clinical harms experienced by patients whose surgery has been delayed due to the COVID-19 pandemic?What tools exist to predict or measure non-clinical harm or negative consequences in individuals facing surgical delays?

Non-clinical harm has been defined by the authors of this paper as being *social harm*, *a reduction in the social aspects of quality of life*, and *economic harm*. Whilst psychological harms (e.g., anxiety and depression) associated with delayed surgical procedures are also highly salient, these are considered to be an inextricable part of the impact of physical, social, and economic harms experienced by the patient and the ongoing presence of the COVID-19 pandemic. Hence, they are not considered separately. In addition, pre-operative anxiety in general is well documented in the literature [6,7] and the review team felt that it may be difficult to isolate any deleterious mental health effects from surgical delay specifically as opposed to pre-operative anxiety more generally. 

## 2. Methods

A rapid scoping review was undertaken in order to demonstrate the nature of existing knowledge on this topic and to identify the key gaps, rather than synthesise evidence in relation to a specific, focused clinical question [8]. The review followed Joanna Briggs Institute (JBI) methodological guidance [9] and is reported in accordance with the PRISMA checklist (scoping review extension) [10]. Scoping reviews seek to identify all types of evidence on a topic. The included evidence is often derived from diverse philosophical paradigms and theoretical underpinnings; hence, scoping reviews are inherently pragmatic, with a focus on producing knowledge that can be actioned through further study. 

### 2.1. Searching

A comprehensive literature search was conducted during October 2020 using a range of approaches: (i) search of 10 electronic databases, including AMED, BNI, CINAHL, EMBASE, EMCARE, MIC, Medline, PsychINFO, Cochrane, and Joanna Briggs Institute; (ii) reference list searching of papers identified for inclusion; (iii) consultation of experts; and (iv) Google Scholar. Google Scholar was used to identify any publications that may not have been indexed by the afore-mentioned search engines, including any grey literature that may have been of relevance. The search strategy was developed with a professional librarian who specialises in conducting literature searches, and a JBI reviewer. The search terms are detailed in Box 1. and the search results are reported in the PRISMA flow diagram [11]. 

Box 1Search terms used during electronic literature search of papers for inclusion.delayed surgery AND optimal stratificationdelayed surgery AND stratification time to treatmentsocial OR economic factors OR financial OR employment delay surgerycancel surgery risk or harm OR stratification OR toolsphenomenology OR experience OR livedtreatment delay surgery elective planned surgery OR scheduled surgeryelective surgical procedureswaiting for surgery

### 2.2. Study Screening and Selection

We included primary and secondary research including: (i) those reporting adult patients’ experiences whilst waiting for surgery in high-income countries with a particular focus on the social, relational, or economic effects; and (ii) studies that utilised assessment tools which measure the extent of these effects. All methodologies and study designs of any date range were considered. Publications specifically about surgery for malignancy and cardiac surgery were excluded because they are a surgical priority. 

All records from the search were imported into a reference-management system and screened according to title and abstract. Potentially relevant papers were then reviewed as full text, with reasons for papers excluded at this stage noted in a table (see Appendix A). As this was a rapid review, this stage was primarily undertaken by one reviewer. However, one third of the full-text articles assessed for eligibility were independently reviewed and verified by a second reviewer to check for consistent and accurate application of the inclusion criteria [10]. 

### 2.3. Data Extraction, Charting and Summary 

As per scoping review guidance, there was no formal assessment of methodological quality. Data on study characteristics (e.g., year, country, surgical condition, methodology, methods, population) were extracted and recorded in an Excel spreadsheet. Data related to the impacts of surgical delay were charted and summarised into five thematic areas. Results are presented narratively using descriptive statistics where appropriate. These steps were undertaken using Excel and were undertaken by the lead author, in discussion with other team members. 

## 3. Results

### 3.1. Overview of Study Characteristics

The search (Figure 1) identified 21 publications, published between 1999 and 2020 and these are detailed in Table 2. Two publications focused on the impact of COVID-19 on surgical waiting times [12] and a possible solution [13]. A full data-extraction table is available (see Appendix A). This includes details regarding the relative contribution (with references) of different studies to the five themes identified.

### 3.2. Themes

The lead author conducted a descriptive content analysis of the 21 publications included in this review to identify findings relevant to our research question. The ‘Population, Context, and Concept’ framework was followed to identify data relevant to the research question, such as behaviours, incidents, beliefs, choices, and emotions [33,34]. Five themes were extracted from the data: impact on employment; impact on social function and leisure activities; impact on patients’ finances; the experience of waiting; and potential patient assessment tools for future utilisation. This latter theme was pre-specified in a deductive manner in order to identify material that was considered potentially suitable for contributing to a surgical delay assessment tool. The first four themes were inductively constructed. The absence of a theoretical framework for this healthcare review, that pragmatically sought to uncover how the non-clinical harms of delayed surgery have been viewed, meant that the deductive theme was discussed by and agreed with all authors during the review design stage.

### 3.3. Impact on Employment

Data on the impact of waiting for surgery on patients’ employment status was reported by 13 publications that had variable findings; one study [12] found that patients’ potential loss of paid work was unknown, yet another review [14] found that longer waits for surgery were associated with a decreased likelihood of returning to work.

Another research team [32] found that amongst people in employment who were waiting for endoscopic sinus surgery (*n* = 18) that 4.8% of time at work was missed and that 34.4% time at work was impaired. Participants in four qualitative papers identified that time off work or adjustments to their work activity was necessary whilst waiting for surgery [17,18,19,20]. Furthermore, reports emerged of longer term negative impacts on career pathway plans [18] and negatively altered relationships with work colleagues [19]. 

Six quantitative publications reported data on the difficulties of maintaining employment by people waiting for surgery, particularly people waiting for joint replacement surgery. Three studies found similar proportions of people resigning from work; 33% (*n* = 71/214) with arthritis [22], and 30% (*n* = 82/278) [31] and 25.7% (*n* = 78/303) [25] of people waiting for a hip or knee replacement. Companies with a small number of employees and no access to occupational health services who could facilitate adjustments to the working environment were more likely to have difficulty retaining staff unable to work whilst waiting for surgery [31]. Pre-surgery sickness absence was also reported amongst some groups; 51% (*n* = 24/47) people waiting for joint replacement surgery [26] and 12% (*n* = 7/55) with gall stones [29]. The same study [29] noted also that 20% (*n* = 13/65) of people waiting for an inguinal hernia repair needed adjustments to their workplace in order to continue working. One paper reported a survey of individuals from five hospitals in the East Midlands (UK) whose surgery was cancelled during the ‘winter pressures’ of 2017/2018 [28]. Of the 339 survey respondents, *n* = 163/399 were of working age (<65 years) and *n* = 111 (68%) were employed. Unplanned working days were lost by 54% (+/−10) of participants. In addition, 33% (*n* = 37/111) of family members needed between one and five days off work to support the patient, totaling 581 days of work lost.

A further paper aiming to report employment-related issues surveyed patients and clinicians on factors which could contribute to the prioritisation of people waiting for surgery [30]. Both the severity of physical symptoms and impact of work had the greatest impact on priority.

### 3.4. Impact on Social Function and Leisure Activities

Data that described the impact of social function and leisure activities was reported by 17 publications, showing that waiting for surgery significantly compromised patients’ leisure activities and activities of daily living [14,20,29,32]. Several publications found that the enforced abandonment of usual roles and activities led to altered relationships with families, friends, and work colleagues, as well as social exclusion [15,16,19,21,25,32]. The likely cause of these effects was identified as either pain or disability, directly leading to disengagement with participants’ social lives [16,17,20,26,29] or the resultant tiredness from poor sleep due to pain or discomfort [32]. Two frequently used health-related quality-of-life (HRQoL) assessment tools were employed by some authors as a research method—the EQ-5D [13,29] and the SF-36 [24,26,27,31]. However, neither the EQ-5D or the HRQoL tools include questions to assess alterations to sleeping patterns which, if impaired, can negatively affect social function among the assessment domains.

Two quantitative papers by the same author [22,23] used an alternative HRQoL assessment instrument, the *Assessment of Quality of Life* (AQoL) [35]. The AQoL actively measures changes in social function such as relationships with others, sleep, and capacity to fulfil family roles, so is a likely more sensitive measure of the impact of a long wait for surgery on social harms. 

Whilst the impact of waiting for surgery on employment and leisure activities was explicitly investigated in many publications, the ability to continue fulfilling roles in the family or as a carer, was reported in only three publications. The first study identified that 6.9% (*n =* 4/58), 3.2% (*n =* 1/31), and 9.8% (*n =* 5/51) of participants waiting for varicose vein, inguinal hernia, and gallstones surgery respectively experienced problems when caring for dependents [30]. The second study found that 53% (*n =* 160/303) of their population waiting for hip-replacement surgery had difficulty when caregiving [25], and the third paper reported patients feeling “useless” because of being unable to undertake usual activities in the home [17]. 

### 3.5. Impact on Patients’ Finances

The financial consequences to patients and their families whilst waiting for extended periods of time for surgery were observed in three publications. This theme therefore focuses specifically on the financial impact of both employment difficulties and the costs associated with additional face-to-face hospital appointments while waiting for surgery. Whilst one research team observed that there were no data identified regarding the effects of surgical cancellation on the patient’s potential economic consequences, e.g., loss of work, sick leave, ability to maintain their housing arrangements [12], other authors reported that 13.3% (*n* = 40/303) of participants waiting for hip or knee surgery experienced a loss in income, although this sum was not quantified or correlated with a specific waiting period [25]. The economic burden experienced by some patients was identified in a survey that found that 48% (*n =* 143/303) incurred additional travel costs for hospital appointments of between GBP 6.50 and 30 [28]. However, the questions asked of patients did not address specific economic issues such as any missed mortgage or rent payments, a need to access food banks, having to prioritise bill payments over food or goods needed for children. 

### 3.6. The Experience of Waiting

Patients’ overall experiences of waiting for their surgery was reported by five publications. A loss of control and agency over the waiting period was a source of distress for many participants in some studies [17,18,20,21]. A few participants reported that the time spent waiting was a positive opportunity to organise and prepare for their upcoming surgery, to plan positive lifestyle changes, and to appreciate their family and friends [18]. Being resigned to waiting and having a fatalistic perspective resulted in greater well-being, although waiting could be challenging for those in paid employment [18]. The experience of waiting may depend on the quality and frequency of communication from patients’ clinical teams, and how people can use their time in the interim.

### 3.7. Potential Patient Assessment Tools for Future Utilisation

The final theme was derived deductively because the authors sought to identify any peer-reviewed and validated patient-assessment tools that could be used in future research to inform the incorporation of non-clinical harm into surgical prioritisation decision making. Three papers reported on the use of two existing published and validated assessment instruments [22,23,32]. A further three publications had developed questions for the purpose of meeting their study’s’ aims and objectives [25,28,30]. 

The first is the *Assessment of Quality of Life* (AQoL) [35], used in two research studies by the same author [22,23]. This assessment tool measures social function, such as the level of assistance needed for personal care and household tasks, social isolation, the capacity to undertake one’s role within the family, and the ability to sleep. The second is the *Work Productivity and Activity Impairment General Health* (WPAI-GH) questionnaire [36], that was used to measure the impact of a given health condition on both work and non-work activity over 7 days’ duration prior to completion [32]. 

Three authors devised their own data collection questions. In the first paper, the *Winter Elective Surgery Cancellation and Psychological impact* (WES-Pi) survey [29] is most closely aligned to the situation brought about by the COVID-19 pandemic. The authors sought to specifically quantify the economic and psychological impact of the cancellation of operations due to winter pressures. In the second paper, researchers investigated the waiting list priority judgements of patients, surgeons, occupational health physicians and general practitioners, using vignettes describing physical symptoms, the psychological distress, social limitations and impairments in work [31]. The authors in the final paper created questions on the acceptability of waiting times, prioritisation of people in pain, and ability to independently undertake activities of daily living [25]. 

## 4. Discussion

The review findings show that, for some patients, the experience of waiting for surgery can include social and economic hardship which might contribute to deleterious effects. It would be reasonable to assume that if a patient has a reduced ability or incapacity to function in their paid employment role, these difficulties would also be transferred to roles or functions in the home. Given the economic consequences for many families who have lost jobs or been furloughed during the COVID-19 pandemic, employment difficulties associated with surgical delay are an important potential non-clinical harm that could impact on the wellbeing of the whole household. The review found that the nature and prevalence of social and economic harm experienced by patients is overall poorly characterised and has certainly not been addressed in the literature to date in the context of a pandemic. These findings resonate with the experiences reported by some patients at the authors’ hospital. There are situations, for example, where patients waiting for stoma reversal surgery cannot leave the house for fear of their stoma bag becoming dislodged and soiling themselves in public, or where the ongoing pain and restricted ability to join in social activities has led to relationship breakdown.

The review identified a limited number of patient assessment tools that could potentially be used to address the current gaps in understanding either in their current format (the *Assessment of Quality of Life* [AQoL] and *the Work Productivity and Activity Impairment General Health* [WPAI-GH] questionnaires) or to be adapted with patient and public involvement (*Winter Elective Surgery Cancellation and Psychological impact* (WES-Pi) survey). Bespoke questions to be incorporated into an assessment tool may be adapted from additional publications [25,30]. Further studies on the impact of surgical delay will need to include a package of work to develop and validate an appropriate assessment tool. 

None of the publications in the review included any consideration of ethical issues related to whether any social, economic, or psychological factors experienced by patients should be formally considered alongside the physical criteria for stratifying the order of waiting for people within their allotted RCS priority group. The authors recognise that many surgeons will be considering how to appropriately incorporate the risk of social harms for patients in their decision-making processes. However, the absence of a transparent structure upon which to base contemporary holistic waiting list prioritisation strategies in the context of post-pandemic management could result in inequalities and a “post-code lottery” of surgical prioritisation. 

This report is subject to limitations. The rapid scoping review methodology is less robust than a formal systematic review and does not accommodate a quality appraisal of the literature selected for inclusion. However, it does allow for a wider body of literature to be considered when there is either a paucity of research or a variety of methodological approaches [37]. 

The social and financial harms experienced by patients who are waiting for extended periods of time for their surgery are important domains for further research. Studies to unpack the granularity of patients’ lived experience in the context of delayed surgery due to the pandemic are required, first to support current patients in this position, and second to inform future pandemic readiness plans. 

## 5. Conclusions

This review was prompted by the need to address surgical cancellations in a UK context and the findings have been considered in relation to UK policy. Nonetheless, the fact that the literature was identified from six different countries suggests that the findings may have a wider applicability. This paper identifies that the experience of a non-clinical harm can be a reality for many patients waiting for surgery. Further research will be crucial for understanding the extent to which non-clinical harms affect patients and carers as well as the wider financial and socio-economic effects. The impact on patients’ nursing care needs due to a decline in physical functioning may also warrant further exploration. In the UK, long waiting lists for surgery are generally presented by the media as a measure of inefficient health care services and insufficient funding. Whilst the current COVID-19 pandemic will afford some respite from that narrative, it is unclear how long that position will hold before the UK health service once again faces criticism. 

## Figures and Tables

**Figure 1 ijerph-19-05542-f001:**
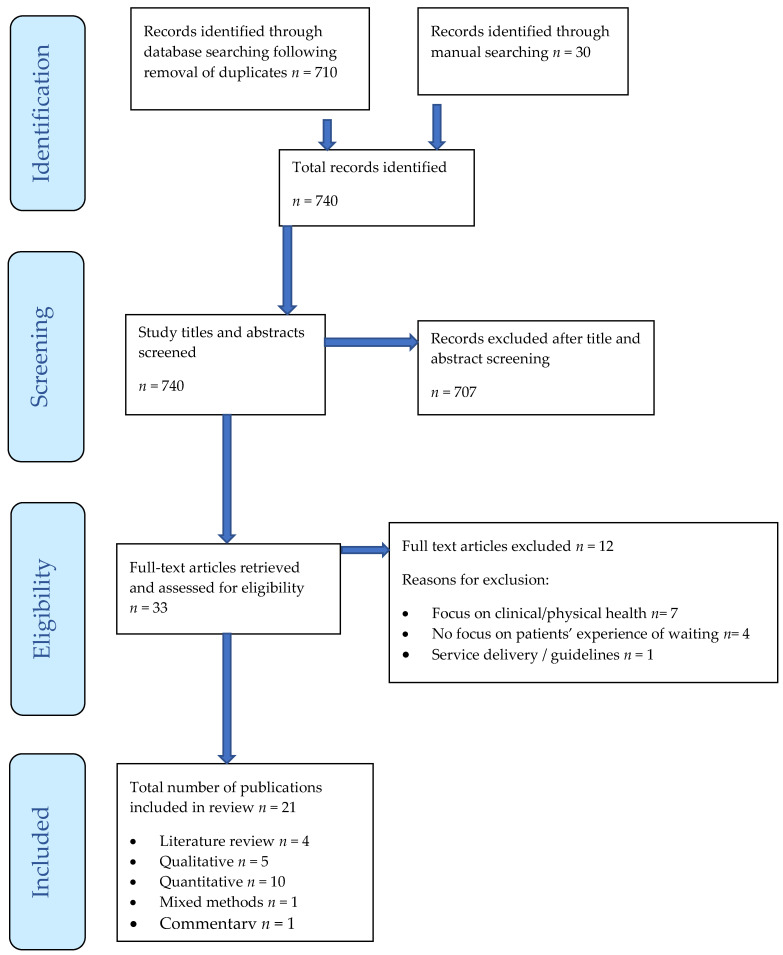
PRISMA Flow Diagram.

**Table 1 ijerph-19-05542-t001:** The Royal College of Surgeons’ (2020) Surgical priority level classifications [3].

Priority Level	Timing of Surgery
1a Emergency	Operation needed within 24 h
1b Urgent	Operation needed with 72 h
2	Surgery that can be deferred for up to 4 weeks
3	Surgery that can be delayed for up to 3 months
4	Surgery that can be delayed for more than 3 months

**Table 2 ijerph-19-05542-t002:** Publications included in the scoping review.

Methodology	Author	Country	Surgical Conditions
Literature reviews *n* = 4	Carr et al., 2009 [14]	Canada	Mixed general surgery
	Morris et al., 2018 [15]	Australia	Orthopedic
	Oudhoff et al., 2004 [16]	Netherlands	Mixed general surgery
	Søreide et al., 2020 [12]	Norway	Mixed general surgery
Qualitative *n* = 5	Carr et al., 2014 and 2017 [17,18]	Canada	Orthopedic and cardiac
	Hilkhuysen et al., 2005 [19]	Netherlands	Mixed general surgery
	Johnson et al., 2014 [20]	UK	Hip replacement
	Sjöling et al., 2005 [21]	Sweden	Hip/knee replacement
Quantitative *n* = 10	Ackerman et al., 2005, 2011 [22,23]	Australia	Hip/knee replacement
	Brownlow et al., 2001 [24]	UK	Hip replacement
	Conner-Spady et al., 2007 [25]	Canada	Hip/knee replacement
	Derrett et al., 1999 [26]	New Zealand	Hip replacement/urology
	Desmeules et al., 2009 [27]	Canada	Knee replacement
	Herrod et al., 2019 [28]	UK	Gall-stones, hernia,
	Oudhoff et al., 2007 and 2007 [29,30]	Netherlands	Mixed general surgery
	Palmer et al., 2005 [31]	UK	Hip/knee replacement
Mixed methods *n* = 1	Tsang et al., 2016 [32]	Canada	Endoscopic sinus surgery
Commentary *n* = 1	de Gorter, 2020 [13]	UK	All elective procedures

## Data Availability

Not applicable.

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
