# Peer review of "Identifying and Understanding the Non-Clinical Impacts of Delayed or Cancelled Surgery in Order to Inform Prioritisation Processes: A Scoping Review"

_ijerph, 2022, doi:10.3390/ijerph19095542_

Round 1

Reviewer 1 Report

This review article addresses the non-clinical impacts of delayed or cancelled surgery that has occurred in the Covid-19 disaster. The global medical response has been slowed by the Covid-19 pandemic, and explicit recognition of the challenges arising from it is essential to address them. Therefore, the review of non-clinical influences presented in this review paper is significant. 

However, when viewed as an academic paper, there are some parts that need to be revised.

In particular, the position of this paper from the theoretical perspective is unclear.

The author(s) should also explain in detail the theoretical background of the research question, such as what problems with existing theories the author recognized and set up the question. 

And what theoretical issues remain in the study's results, and what possible directions for future research should also be explicitly and holistically discussed.

Below are detailed comments tied to the page number and line number.

---------------

Page 1, 12: 
"The ways ... is currently unclear." should be "The ways ... are currently unclear."?

Page 2, 72: 
The questions for this study are presented, but it is unclear what theoretical background the perspective is positioned in. The author(s) should also clarify how the research question is positioned from a theoretical perspective. For example, this requires clarification of the positioning of the research gap from a theoretical perspective.

Page 3, 97: 
There was no description of how Google Scholar was used. The author(s) should clearly state how Google Scholar was used.

Page 5, 130: 
There is an extra parenthesis (...waiting times ([13]...)that should be removed.

Page 5, 134: 
In general, figures and tables should be numbered.

Page 6, 182: 

"... whist ..." should be "... whilst ..."

Page 8, 273: 
It is expected that the section of the Discussion will include suggestions that have readable implications for researchers and practitioners. In particular, review papers typically discuss perspectives on what research issues remain in the subject area and what research directions are possible. Thus, the author(s) should also clearly present direction(s) for future research based on the results of this review.

---------------

Author Response

Reviewer 1.

The authors thank reviewer 1 for their detailed consideration of our manuscript. We have responded to their comments as below;

This review article addresses the non-clinical impacts of delayed or cancelled surgery that has occurred in the Covid-19 disaster. The global medical response has been slowed by the Covid-19 pandemic, and explicit recognition of the challenges arising from it is essential to address them. Therefore, the review of non-clinical influences presented in this review paper is significant. 

However, when viewed as an academic paper, there are some parts that need to be revised.

In particular, the position of this paper from the theoretical perspective is unclear.

The author(s) should also explain in detail the theoretical background of the research question, such as what problems with existing theories the author recognized and set up the question. 

And what theoretical issues remain in the study's results, and what possible directions for future research should also be explicitly and holistically discussed.

Scoping literature reviews in nursing are not always underpinned by an academic theoretical perspective, but rather a pragmatic problem-solving approach to a contemporary problem. However we realise that many readers may wish us to be explicit about this so have added the following sentences to the Themes section, page 186:

“The absence of a theoretical framework for this healthcare review, that pragmatically sought to uncover how the non-clinical harms of delayed surgery have been viewed, meant that the deductive theme was discussed by and agreed with all authors during the review design stage”

In addition, in the Methods section (lines 96-99) we have added the following clarification

Scoping reviews seek to identify all types of evidence on a topic. The included evidence is often derived from diverse philosophical paradigms and theoretical underpinnings, hence scoping reviews are inherently pragmatic with a focus on producing knowledge that can be actioned through further study.

Below are detailed comments tied to the page number and line number.

---------------

Page 1, 12: 
"The ways ... is currently unclear." should be "The ways ... are currently unclear."?

Changed, see line 12

Page 2, 72: 
The questions for this study are presented, but it is unclear what theoretical background the perspective is positioned in. The author(s) should also clarify how the research question is positioned from a theoretical perspective. For example, this requires clarification of the positioning of the research gap from a theoretical perspective.

Line 73, text added: . A pragmatic, problem-solving approach was considered necessary by the research team to gather information to use to improve patient care.

Page 3, 97: 
There was no description of how Google Scholar was used. The author(s) should clearly state how Google Scholar was used.

Text added, line 102: Google Scholar was used to identify any publications that may not have been indexed by the afore-mentioned search engines, including any grey literature that may have been of relevance.

Page 5, 130: 
There is an extra parenthesis (...waiting times ([13]...)that should be removed.

Parenthesis removed.

Page 5, 134: 
In general, figures and tables should be numbered.

Figure 1. label added to Prisma diagram line 148

Page 6, 182: 

"... whist ..." should be "... whilst ..."

Changed line 210

Page 8, 273: 
It is expected that the section of the Discussion will include suggestions that have readable implications for researchers and practitioners. In particular, review papers typically discuss perspectives on what research issues remain in the subject area and what research directions are possible. Thus, the author(s) should also clearly present direction(s) for future research based on the results of this review.

The following text has been added at line 339;

Further studies on the impact of surgical delay will need to include a package of work to develop and validate an appropriate assessment tool.

The following text has been added to line 349:

The social and financial harms experienced by patients who are waiting for extended periods of time for their surgery are important domains for further research. Studies to unpack the granularity of patients’ lived experience in the context of delayed surgery due to a virus pandemic are required first to support current patients in this position, and second to inform future pandemic readiness plans. 

Reviewer 2 Report

I would like to thank the authors for their work.

This is an interesting paper, on a very timely and underestimated topic.

This manuscript aims to identify material to inform the construction of a contemporary assessment tool to aid surgical priority decision making in the aftermath of the Covid-19 pandemic.

The methodology is robust and the article has a high scientific soundness.

The results are clear and consistent with the main question, and well-deemed with thea available literature.

The main question of this work is well addressed by the research conducted, the topic is certainly relevant in literature, adding new perspectives for the scientific community, even if it is a scoping review.

This work can be continued with a systematic review based on the conclusions.

The manuscript is well written and easy to read.

The results are clear and consistent with the main question, and well-deemed with the available literature.

Author Response

Reviewer 2.

The authors thank reviewer 2 for their detailed consideration of our manuscript.

Reviewer 3 Report

I would like to congratulate the authors for what I believe is a very interesting, well-conducted and pertinent work. In fact, I have only a few notes:

  • it is unclear how the themes were constructed. The authors mention that "four themes were inductively constructed, and the fifth theme contains deductively identified material"; however, it is not specified what evidence was used as a starting point for theme definition. Were inductive themes based merely on the researchers' opinions and expectations or was there a more unbiased process underlying theme definition, using, for example, previous sources that analysed non-clinical impacts of delayed or cancelled surgeries?
  • the discussion section, though well-written and thought-provoking, is a bit short. I would like to read more about the authors' insight on the subject, and what next steps they deem to be best for furthering research in this area and moving towards a better system for surgery prioritization.

Author Response

Reviewer 3.

The authors thank reviewer 3 for their detailed consideration of our manuscript. We have responded to their comments as below;

I would like to congratulate the authors for what I believe is a very interesting, well-conducted and pertinent work. In fact, I have only a few notes:

  • it is unclear how the themes were constructed. The authors mention that "four themes were inductively constructed, and the fifth theme contains deductively identified material"; however, it is not specified what evidence was used as a starting point for theme definition. Were inductive themes based merely on the researchers' opinions and expectations or was there a more unbiased process underlying theme definition, using, for example, previous sources that analysed non-clinical impacts of delayed or cancelled surgeries?

Changes have been made to the paragraph titled ‘Themes’ from line 181:

The lead author conducted a descriptive content analysis of the 21 publications included in this review to identify findings relevant to our research question. The ‘Population, Context, and Concept’  framework  was followed to identify data relevant to the research question, such as behaviors, incidents, beliefs, choices and emotions (Gale et al., 2013; Peters et al., 2021). Four themes were inductively constructed, and the fifth theme contains deductively identified material that could be developed for use as an assessment tool. The absence of a theoretical framework for this healthcare review, that pragmatically sought to uncover how the non-clinical harms of delayed surgery have been viewed, meant that the deductive theme was discussed by and agreed with all authors during the review design stage.

  • the discussion section, though well-written and thought-provoking, is a bit short. I would like to read more about the authors' insight on the subject, and what next steps they deem to be best for furthering research in this area and moving towards a better system for surgery prioritization.

Insights have been added into line 323:

These findings resonate with the experiences reported by some patients at the authors’ hospital. There are situations, for example, where patients waiting for stoma reversal surgery cannot leave the house for fear of their stoma bag becoming dislodged and soiling themselves in public, or where the ongoing pain and restricted ability to join in social activities has led to relationship breakdown.

And additional text about future research has been added to line 349:

The social and financial harms experienced by patients who are waiting for extended periods of time for their surgery are important domains for further research. Studies to unpack the granularity of patients’ lived experience in the context of delayed surgery due to a virus pandemic are required first to support current patients in this position, and second to inform future pandemic readiness plans. 

Round 2

Reviewer 1 Report

Thank you for making the corrections based on review comments. The analysis procedure is now clearer. 

But there are still several corrections to be made in reading the manuscript.

Below are the comments tied to the page and line numbers.

---------------------

% Page 6, line 183: 3. Results/Themes: 

At this stage here, it is not clear what the intent of the explanation of this paragraph is, as there is no explanation of what specific themes were extracted. For instance, first of all, the author(s) should be to present the big picture regarding the five classification results in this study, so that the reader can grasp an overview of the results.

% Page 6, line 196: 3. Results/Impact on employment/Par.1
This paragraph does not explain what this theme is. In the main text, the author(s) should first explain the theme here and then introduce each of the paper's content. This applies equally to the description of the subsequent theme's paragraph. 

% Page 7, line 225: 3. Results/Impact on employment/Par.3

The author(s) should explain what the "mean" here represents.

% Page 7, line 237: 3. Results/Impact on social function.../Par.1

There are overlapping round brackets. Please check the journal guidelines for citation notations. There are several same points in the manuscript. 

---------------------

Author Response

Thank you very much for your close reading of this manuscript and identifying further points to address.

% Page 6, line 183: 3. Results/Themes: 

At this stage here, it is not clear what the intent of the explanation of this paragraph is, as there is no explanation of what specific themes were extracted. For instance, first of all, the author(s) should be to present the big picture regarding the five classification results in this study, so that the reader can grasp an overview of the results.

Sentence added in blue ink.

% Page 6, line 196: 3. Results/Impact on employment/Par.1
This paragraph does not explain what this theme is. In the main text, the author(s) should first explain the theme here and then introduce each of the paper's content. This applies equally to the description of the subsequent theme's paragraph. 

Sentence added in blue ink to each of the five themes.

% Page 7, line 225: 3. Results/Impact on employment/Par.3

The author(s) should explain what the "mean" here represents.

This sentence has been re-worded for clarity.

% Page 7, line 237: 3. Results/Impact on social function.../Par.1

There are overlapping round brackets. Please check the journal guidelines for citation notations. There are several same points in the manuscript. 

I have checked the manuscript for any further anomalies with the citations.